# Unveiling the Multitarget Potential of a Rare Caffeoyl Ester from *Artemisia capillaris* for Diabetes Mellitus: An Integrated In Vitro and In Silico Study

**DOI:** 10.3390/ijms26031286

**Published:** 2025-02-02

**Authors:** Md. Nurul Islam, Manh Tuan Ha, Byung-Sun Min, Jae Sue Choi, Hyun Ah Jung

**Affiliations:** 1Department of Pharmacy, Faculty of Life Science, Mawlana Bhashani Science and Technology University, Tangail 1902, Bangladesh; nurul205@mbstu.ac.bd; 2Drug Research and Development Center, College of Pharmacy, Daegu Catholic University, Gyeongbuk 38430, Republic of Korea; 3Department of Food and Life Science, Pukyong National University, Busan 48513, Republic of Korea; choijs@pknu.ac.kr; 4Department of Food Science and Human Nutrition, Jeonbuk National University, Jeonju 54896, Republic of Korea

**Keywords:** *Artemisia capillaris*, 4-*O*-caffeoyl-2-*C*-methyl-d-threonic acid, anti-diabetic, protein tyrosine phosphatase 1B, aldose reductase

## Abstract

As a part of our ongoing search for bioactive constituents of *Artemisia capillaris*, we isolated 4-*O*-caffeoyl-2-*C*-methyl-d-threonic acid (PPT-14). This is a rare caffeic acid ester derivative that is reported here for the first time in the Artemisia species, which is the third occurrence in any plant species worldwide. In this study, we evaluated the anti-diabetic potential of PPT-14 using in vitro and in silico approaches. PPT-14 demonstrated significant inhibitory activity against two crucial enzymes linked to diabetes progression and complications: protein tyrosine phosphatase 1B (PTP1B) and aldose reductase (AR). These had IC_50_ values of 64.92 and 19.50 µM, respectively. Additionally, PPT-14 exhibited free radical scavenging activity with 2,2-diphenyl-2-picrylhydrazyl (IC_50_ 14.46 µM). Molecular docking and 200 ns molecular dynamics simulations confirmed that there were stable binding interactions with the key residues of PTP1B and AR, highlighting strong affinity and dynamic stability. Pharmacokinetic analyses revealed favorable water solubility, adherence to Lipinski’s Rule of Five, and minimal interactions with cytochrome P450 enzymes, indicating the drug-like potential of PPT-14. Toxicity studies confirmed its safety profile, showing no genotoxicity, hepatotoxicity, or significant toxicity risks, with an acceptable oral LD_50_ value of 2.984 mol/kg. These findings suggest that PPT-14 could be a promising multitarget lead compound for ameliorating diabetes and its associated complications.

## 1. Introduction

Diabetes is a complex, chronic, and progressively worsening metabolic disorder, caused by either an absolute or relative deficiency of insulin or by reduced insulin activity. It results in hyperglycemia, with a fasting blood glucose level of ≥126 mg/dL [1]. The disease is characterized by severe complications, such as nephropathy, neuropathy, and retinopathy, which can damage vital organs, including the pancreas, heart, liver, adipose tissue, and kidneys, making it a leading cause of death worldwide [2]. Although the exact cause of diabetes remains uncertain, scientific evidence indicates that the production of free radicals and the reduced ability to neutralize them significantly contribute to the progression of the disease. These factors are particularly critical in the onset of diabetes-related complications. Chronic hyperglycemia leads to the generation of free radicals, which can damage cellular structures, including DNA, proteins, and lipids, thereby disrupting normal physiological functions. Additionally, it promotes the formation of advanced glycation end products that directly affect cells, triggering inflammation and oxidative stress [3]. Because persistent hyperglycemia is the leading cause of diabetic complications, the primary aim of diabetes management is to achieve and maintain normal blood glucose levels through a combination of anti-diabetic medications and lifestyle changes. Several treatment strategies are available for diabetic patients, including synthetic drugs and medicinal plant products from natural sources. The currently available synthetic drugs include thiazolidinediones, insulin secretagogues, glucagon-like peptide 1 (GLP-1) agonists, dipeptidyl peptidase IV (DPP-IV) inhibitors, α-glucosidase inhibitors, and peroxisome proliferator-activated receptor (PPAR)-γ agonists [4]. Although many of these conventional anti-diabetic drugs are powerful and effective, they are not sufficient to maintain optimal treatment for all patients, especially as the disease advances over time [5]. This situation is further exacerbated by the range of side effects of modern synthetic drugs, such as gastrointestinal disorders, hypoglycemia, and hepatotoxicity [6,7]. Therefore, to achieve normoglycemia and manage the variety of complications that afflict patients with diabetes, the development of a comprehensive and multi-dimensional treatment approach with fewer side effects is essential. In addition to prescribed medications, most diabetic patients in developing countries rely on complementary or natural medicinal products known for their anti-diabetic properties. Due to their rich content of diverse phytochemicals, medicinal plants can contribute to diabetes management by influencing multiple pathways that are involved in the development and progression of diabetes-related complications. Recently, the use of medicinal plants and their derivatives as interventions for diabetes mellitus has increased significantly because of their minimal side effects, the synergistic action of their phytochemicals, and their affordability [8,9].

The *Artemisia* genus, belonging to the family Asteraceae, is considered one of the most widely distributed genera across the world. With a rich diversity comprising over 400 species, including grasses, shrubs, and some trees, it occupies many ecological niches across the planet [10]. *Artemisia* plants have attracted widespread attention and received extensive use in traditional medicine due to their varied chemical compositions, which can include lignans, sesquiterpenoids, flavonoids, coumarins, glycosides, caffeoylquinic acids, sterols, and polyacetylenes, resulting in a wide range of pharmacological effects [10]. Among *Artemisia* species, *A. capillaris*, which is known as Haninjin and Injinho in Korea, is commonly found to be growing in sandy areas along the Korean coastline. This plant has long been used in traditional medicine for the treatment of many diseases, including hepatitis, jaundice, fatty liver, and bilious disorder. Infusions of the buds, stems, and leaves of *A. capillaris.* have been used in traditional Chinese medicine, primarily as choleric, anti-inflammatory, antipyretic, and diuretic agents for treating epidemic hepatitis [11]. *Artemisia capillaris* has been reported to exhibit a broad range of pharmacological and biological activities, including antioxidant, cytoprotective, hepatoprotective, anti-inflammatory, antimicrobial, anticancer, anti-obesity, and choleretic effects [12,13,14,15,16,17,18,19]. *Artemisia capillaris* is rich in several phytochemicals, including flavonoids, coumarins, chromones, mono- and sesquiterpenes, polyacetylenes, and dicaffeoyl quinic acid derivatives, elements which are likely linked to its potential bioactivity [16,20,21,22,23]. In recent years, numerous studies have highlighted the promising anti-diabetic potential of *A. capillaris* and its isolated phytochemicals, particularly flavonoids, coumarins, polyacetylenes, and caffeoyl quinic acid derivatives. These compounds target multiple aspects of diabetes and related complications, including the inhibition of aldose reductase (AR), protein tyrosine phosphatase 1B (PTP1B), and α-glucosidase [21,24,25]. As part of our ongoing research on novel bioactive constituents from *A. capillaris*, we isolated a methanol (MeOH)-soluble rare caffeic acid ester derivative that we identified as 4-*O*-caffeoyl-2-*C*-methyl-d-threonic acid (PPT-14). The present study was designed to evaluate the anti-diabetic potential of PPT-14 using in vitro and in silico studies.

## 2. Results

### 2.1. Isolation of PPT-14

Repeated column chromatography of the *n*-BuOH fraction of *A. capillaris* yielded PPT-14 (Figure 1). The structure of the isolated compound was confirmed by ^1^H and ^13^C NMR spectroscopy as well as via comparison with the published literature [26,27].

PPT-14: Brown solid. ^1^H NMR (500 MHz, D_2_O) *δ* (ppm): 7.44 (1H, d, *J* = 15.9 Hz, H-7′), 7.02 (1H, s, H-2′), 6.94 (1H, d, *J* = 7.8 Hz, H-6′), 6.83 (1H, d, *J* = 7.8 Hz, H-5′), 6.19 (1H, d, *J* = 15.9 Hz, H-8′), 4.21 (2H, d, *J* = 5.7 Hz, H-4), 4.08 (1H, t, *J* = 5.7 Hz, H-3), 1.41 (3H, s, H-5). ^13^C NMR (125 MHz, D_2_O) *δ* (ppm): 180.6 (C-1), 169.4 (C-9′), 147.3 (C-4′), 146.0 (C-7′), 144.4 (C-3′), 126.8 (C-1′), 122.7 (C-6′), 116.2 (C-5′), 115.1 (C-2′), 114.1 (C-8′), 76.9 (C-2), 73.6 (C-3), 65.5 (C-4), 22.6 (C-5).

### 2.2. Inhibitory Activity of PPT-14 Against α-Glucosidase, PTP1B, and HRAR Enzymes

The inhibitory effect of PPT-14 against α-glucosidase was determined by using pNPG as a substrate. PPT-14 did not exhibit any inhibitory activity against α-glucosidase with the tested concentration (Table 1). The inhibitory effect of PPT-14 against PTP1B was determined using p-NPP as a substrate. As shown in the Table 1, PPT-14 showed promising PTP1B inhibitory activity with an IC_50_ value of 64.92 ± 0.34 μM compared to the positive control ursolic acid, which had an IC_50_ value of 6.11 ± 0.25 μM. The HRAR-inhibitory activity of PPT-14 was evaluated using Renilla luciferase assay reagent (RLAR) and dl-glyceraldehyde as a substrate. PPT-14 showed very strong RLAR inhibitory activity with an IC_50_ value of 19.50 ± 0.51 compared to the positive control quercetin, which had an IC_50_ value of 3.20 ± 0.23 μM (Table 1).

### 2.3. Free Radical Scavenging Activity of PPT-14

PPT-14 exhibited very strong DPPH free radical scavenging activity with an IC_50_ value of 14.46 µM compared to the positive control ascorbic acid, which had an IC_50_ value of 31.79 µM (Table 1).

### 2.4. Molecular Docking of PPT-14 Against PTP1B and AR

In this study, we investigated the binding affinity and amino acid interactions of PPT-14 and ursolic acid with the PTP1B (PDB ID: 1NNY). The results, summarized in Table 2, highlight the key hydrogen bonding and hydrophobic interactions contributing to ligand binding and stability. The binding affinity of PPT-14 to 1NNY was calculated to be −7.0 kcal/mol, while ursolic acid exhibited a stronger binding affinity of −8.7 kcal/mol. Hydrogen bonding played a significant role in stabilizing the ligand–protein complex. PPT-14 was observed to form a total of seven hydrogen bonds with the residues Asp48, Gly183 (two bonds), Arg221, Thr263, and Gln266 (two bonds). These had bond distances of 1.86–2.82 Å, suggesting the existence of strong polar interactions at these sites (Figure 2A). The hydrophobic interactions of PPT-14 with 1NNY involved residues Trp179 (2.23 Å), Gly220 (3.45 Å), and Ala217 (3.79 Å). Ursolic acid formed fewer hydrogen bonds than PPT-14, interacting with Asp48 (2.80 Å), Trp179 (2.56 Å), and Arg221 (2.40 Å). In contrast, ursolic acid demonstrated a greater number of hydrophobic interactions than PPT-14 (Figure 2B). Notably, interactions were observed with Tyr46 (multiple bonds at 3.98, 3.99, and 4.09 Å), Val49 (4.09 Å), Phe182 (5.42 Å), and Ala217 (3.97 Å). The binding affinity of PPT-14 with 1EL3 was determined to be −9.1 kcal/mol, whereas epalrestat showed a slightly higher binding affinity of −9.6 kcal/mol. PPT-14 formed seven hydrogen bonds with the protein, interacting with key residues such as Thr19 (2.35 Å), Trp20 (2.29 Å), Asn160 (1.84 and 2.88 Å), Tyr209 (3.05 Å), Ser214 (2.99 Å), and Asp216 (3.02 Å). Notably, the short bond distance of 1.84 Å between PPT-14 and Asn160 indicated that there was a particularly strong polar interaction. Hydrophobic interactions also played a significant role in stabilizing both ligand–protein complexes. PPT-14 established hydrophobic interactions with Lys21 (1.11 Å), Tyr209 (3.88 Å), Ser210 (3.35 Å), Pro211 (3.56 Å), and Cys298 (4.72 Å) (Figure 3A). In contrast, epalrestat formed a single hydrogen bond with Cys298 (2.75 Å) but exhibited a greater number of hydrophobic interactions, likely contributing to its slightly stronger binding affinity. These hydrophobic interactions involved the residues Trp20 (4.15 and 6.14 Å), Tyr48 (4.69 Å), Trp48 (5.79 Å), His110 (4.87 and 5.07 Å), Tyr209 (3.86 Å), Ile260 (5.35 Å), and Cys298 (5.09 Å) (Figure 3B). The extensive involvement of multiple residues, particularly aromatic residues such as Trp20, Trp48, and Tyr209, suggested that epalrestat formed a robust network of van der Waals interactions within the binding pocket.

### 2.5. Molecular Dynamics (MD) Simulation

MD simulations were performed in real time to assess the stability of protein–ligand complexes under conditions that mimicked the human physiological environment. These simulations also offered insights into conformational changes in the protein complexes within the computational framework. For the best understanding of complex stability, PPT-14–protein complexes along with the protein–reference complex were subjected to 200 ns simulations. This allowed us to view PPT-14 as a potent inhibitor of PTP1B and AR.

#### 2.5.1. Root-Mean-Square Deviation and Fluctuation (RMSD and RMSF), Radius of Gyration (R_g_), Solvent-Accessible Surface Area (SASA), and Molecular Interactions of PPT-14 Against PTP1B

The analysis of the root-mean-square deviation (RMSD) of the PPT-14–PTP1B and ursolic acid–PTP1B complexes and the apo protein demonstrated stability within acceptable ranges throughout the simulations (Figure 4A). The apo protein exhibited the most stable RMSD values. For the ursolic acid–PTP1B complex, deviations were noted after 180 ns but these gradually decreased by the end of the simulations. In contrast, the RMSD values for the PPT-14–PTP1B complexes showed no significant deviations. The root-mean-square fluctuation (RMSF) values for the ursolic acid–PTP1B complex were similar with the apo protein (Figure 4B). The PPT-14–PTP1B complex had higher fluctuations at residues 24–33, 147–150, 195–206, and 232–242, while at residues between 21 and 31 and 111 and 119, higher fluctuations were observed for the ursolic acid–PTP1B complex. The PPT-14–PTP1B and ursolic acid–PTP1B complexes and the apo protein exhibited nearly identical radii of gyration (R_g_) profiles (Figure 4C). The PPT-14–PTP1B complexes exhibited lower solvent-accessible surface area (SASA) values, whereas the ursolic acid–PTP1B complex displayed SASA values comparable to those of the apo protein (Figure 4D).

Figure 5 presents the interactions in 200 ns of MD simulations in the PPT-14–PTP1B complex and the reference drug ursolic acid–PTP1B complex. PPT-14 formed hydrogen bonds with several amino acid residues of PTP1B, including Ser29, Phe30, Asp48, Ser216, Arg254, and Gln262, whereas it exhibited hydrophobic interactions with the residues of Tyr46, Val49, and Ile219 (Figure 5A). In contrast, only two hydrogen bonds at residues of Asp48, Trp179, Gly183, and Gln262 were observed with ursolic acid, along with several hydrophobic and ionic interactions at different residues (Figure 5B).

#### 2.5.2. RMSD, RMSF, R_g_, SASA and Molecular Interactions of PPT-14 Against AR

MD simulations were performed to compare the effect of the presence of a peptide on the stability of the three complexes: PPT-14–AR, epalrestat–AR, and the AR apo. The analysis of the RMSD results showed that PPT-14–AR was more stable than epalrestat–AR and the apo protein (Figure 6A). The RMSD values of epalrestat–AR complex continued to deviate the most after 100 ns of MD simulations, while for the PPT-14–AR complex, a slight deviation was observed after 94 ns, which became stable after 115 ns. Furthermore, the apo protein exhibited deviations after 160 ns and its RMSD values remained high afterward. Nevertheless, the deviations were within the acceptable range (<3 Å). In this research, the RMSF values were measured to detect alterations in amino acids and protein structures because of the attachment of small molecules with a specific target protein and their residues within the protein. The RMSF graph analysis indicated that the RMSF values of the compound-protein complexes fluctuated at residues 20–28, 122–130, 170–178, 212–227, 261–270, and 285–310 (Figure 6B). The largest fluctuations were observed between residues 212 and 227 for each complex. The apo protein fluctuated the most at residues of 285–310. The PPT-14–AR complex displayed more stable R_g_ profiles compared to the apo protein (Figure 6C). The R_g_ values for the epalrestat–AR complex were consistent with those of the apo protein up to 164 ns, after which they began to decrease. A similar trend was observed in the SASA profiles, in which the epalrestat–AR complex closely resembled the apo protein up to 164 ns (Figure 6D). The SASA values for the PPT-14–AR complex were nearly identical up to 106 ns, after which they decreased.

The PPT-14–AR and reference drug epalrestat–AR interactions were thoroughly analyzed through 200 ns of MD simulations. PPT-14 formed hydrogen bonds with several amino acid residues of AR, including Lys77, Ser159, Asn160, and Lys262, whereas it exhibited both hydrophobic and hydrogen bonding interactions with the residues of His110 and Ile260 (Figure 7A). In contrast, only two hydrogen bonds were observed for residues of Trp111 and Cys298 with epalrestat, along with several hydrophobic interactions for the residues of Tyr48, Trp79, Tyr209, and Ile260 (Figure 7B).

### 2.6. Pharmacokinetic Properties and Toxicity Studies of PPT-14

Pharmacokinetic analysis revealed that PPT-14 demonstrated an acceptable absorption profile with poor lipophilicity but favorable water solubility (Table 3 and Table 4). The compound was found to be impermeable to the blood–brain barrier (BBB), indicating minimal potential for neurotoxicity. PPT-14 was neither a substrate nor an inhibitor of CYP3A4 or CYP2D6, suggesting that there was no direct interaction with these key metabolic enzymes. PPT-14 adhered to Lipinski’s Rule of Five without violations, indicating strong drug-like properties, though one PAINS violation was identified that was caused by the catechol A group. Despite this, the compound displayed lead-like characteristics. Toxicity assessments showed that there was no genotoxicity or hepatotoxicity, and the compound did not exhibit minnow toxicity (Table 3). The oral rat acute toxicity (LD_50_) value was calculated to be 2.984 mol/kg, suggesting a low acute toxicity risk.

## 3. Discussion

Diabetes is a chronic and debilitating disease that profoundly impacts the lives and health of individuals, families, and communities worldwide. It ranks among the top ten leading causes of death in adults, causing an estimated four million deaths globally in 2017. Diabetes is a widespread and growing public health issue, with 536.6 million diagnosed cases globally in 2021, and projections indicating that this number will rise to 783.2 million by 2045 [28]. The ratio of people with type 2 diabetes is increasing rapidly worldwide, with the majority of cases occurring in underdeveloped and developing countries. Chronic high plasma glucose levels in diabetes can result in major complications, including coronary heart disease, kidney and nerve damage, retinopathy, and oral health problems. They also elevate the risk of cancer, liver disease, infections, and cognitive and emotional disorders [29,30]. Although several synthetic oral medications have been developed to manage diabetes, controlling the diabetes-related complications without side effects remains a challenge. This has sparked growing scientific interest in traditional remedies, with increased research in complementary drugs and natural therapies, driven by the search for more effective treatments with fewer side effects [31,32]. As part of our ongoing exploration of anti-diabetic phytochemicals from *A. capillaris*, an important member of the *Artemisia* genus rich in a variety of bioactive compounds, we successfully isolated a caffeic acid ester derivative identified as PPT-14. This compound was characterized by using a combination of ^13^C and ^1^H NMR chemical shifts, and its identity was confirmed by comparisons with the published literature. Interestingly, to the best of our knowledge, this is the first report of this compound from *Artemisia* species and only the third occurrence of this phytoconstituent in any plant species worldwide. We investigated the antioxidant potential of PPT-14 using the DPPH free radical scavenging assay, which measures the compound’s ability to neutralize free radicals. Interestingly, PPT-14 was found to be a potent antioxidant, with an IC_50_ value of 4.5 µg/mL, compared to the positive control, ascorbic acid, which exhibited an IC_50_ value of 5.6 µg/mL. Additionally, we assessed the anti-diabetic activity of PPT-14 by targeting three key enzymes, α-glucosidase, PTP1B, and AR, which are closely linked to the development and progression of diabetes-related complications. By evaluating its effects on these enzymes, we aimed to better understand the potential of PPT-14 as a therapeutic agent for managing diabetes and its associated metabolic disorders.

α-Glucosidase is a crucial enzyme in carbohydrate metabolism, playing a pivotal role in regulating postprandial glucose levels. Its activity is closely linked to the pathophysiology of diabetes, particularly type 2 diabetes mellitus. However, in this study, PPT-14 did not exhibit any α-glucosidase inhibitory activity at the tested concentrations. Protein tyrosine phosphatases (PTPs) are an essential superfamily of signaling enzymes that regulate a variety of cellular functions, including cell proliferation, differentiation, adhesion, and motility. These activities are crucial for maintaining normal cellular operations and enabling cells to respond effectively to environmental signals throughout their lifecycle [33,34]. PTP1B is a prominent member of the PTP superfamily, is ubiquitously expressed, and exists as a soluble cytosolic protein with a molecular weight of approximately 50 kDa. PTP1B plays a key role in the regulation of insulin action by the dephosphorylation of the activated, autophosphorylated insulin receptor (IR) and downstream substrate proteins such as IRS-1/2. Therefore, PTP1B acts as a negative regulator of the IR signaling pathway, as evidenced by studies showing that mice deficient in the PTP1B gene exhibit enhanced insulin sensitivity and a significantly lower risk of developing obesity and diabetes [35]. It also interacts with and dephosphorylates Jak2, the primary tyrosine kinase responsible for initiating leptin receptor signaling. An accumulation of evidence demonstrates that the overexpression of PTP1B suppresses leptin receptor signaling. Conversely, mice with either the global or neuron-specific deletion of PTP1B exhibit a lean phenotype, heightened sensitivity to leptin, and resistance to diet-induced obesity [36,37]. Therefore, the reversible modulation of insulin, as well as leptin receptor phosphorylation and signaling, holds significant clinical promise, highlighting the potential of PTP1B inhibition to reduce insulin resistance, restore glycemic balance, and target both type 2 diabetes and obesity [36,38]. PTP1B, a 50 kDa protein composed of 435 amino acid residues, features distinct functional regions: an N-terminal catalytic domain (residues 1–300), a regulatory region (residues 300–400), and a C-terminal membrane localization domain (residues 400–435). Structurally, the protein includes 8 α-helices and 11 β-strands. Key structural elements involved in phosphotyrosine dephosphorylation include the R loop (Val113–Ser118), the lysine loop (Leu119–Cys121), the WPD loop (Thr177–Pro185), the S loop (Ser201–Gly209), and the Q loop (Ile261–Gln262), as well as the α3 helix (Glu186–Glu200), the α6 helix (Ala264–Ile281), and the α7 helix (Val287–Ser295) [39]. In this study, PPT-14 significantly inhibited PTP1B, with an IC_50_ value of 64.92 µM, compared to the positive control ursolic acid, which had an IC_50_ value of 6.11 µM. After obtaining promising results from the in vitro experiments, we performed in silico studies of PPT-14 against PTP1B enzyme (PDB ID: 1NNY) using molecular docking and MD simulations. We also compared our data with ursolic acid, a well-known PTP1B enzyme inhibitor. MD simulation, a dynamic trajectory analysis technique used in the estimation of protein and ligand biomolecular interactions, is crucial to the discovery of novel drugs. In this analysis, the Maestro 2020.4 program was used to conduct MD simulations for selected protein–ligand complexes for 200 ns with the predetermined physicochemical parameters. Protein structural reliability and conformational changes in the target protein were measured using an RMSD model.

The stability and dynamic behavior of the PPT-14–PTP1B and ursolic acid–PTP1B complexes and the apo protein were thoroughly evaluated by analyzing the RMSD, RMSF, R_g_, and SASA. The RMSD analysis (Figure 4A) revealed that all the systems maintained stability within acceptable ranges throughout the simulations. Among these, the apo protein exhibited the most consistent RMSD values, indicating its high structural stability. In the case of the ursolic acid–PTP1B complex, deviations were observed after 180 ns. However, these fluctuations stabilized by the end of the simulation, suggesting the system’s ability to regain stability. Notably, the PPT-14–PTP1B complex demonstrated no significant deviations in RMSD, further confirming its robust stability during the simulation period. RMSF analysis (Figure 4B) highlighted localized fluctuations in specific regions of the protein. While the RMSF values for the ursolic acid–PTP1B complex were nearly identical to those of the apo protein, the PPT-14–PTP1B complex exhibited higher fluctuations in the regions spanning residues 24–33, 147–150, 195–206, and 232–242. These fluctuations may correspond to areas involved in ligand interactions or flexible loop regions. Similarly, the co-crystallized ligand–PTP1B complex showed increased fluctuations at residues 21–31 and 111–119 compared to the other systems, potentially indicating regions of reduced stability or increased flexibility upon ligand binding. The R_g_ profiles (Figure 4C) for the PPT-14–PTP1B, ursolic acid–PTP1B complexes, and the apo protein were nearly identical, suggesting the comparable compactness of these systems. In contrast, differences between the complexes were observed in the SASA analysis (Figure 4D), which provided further insights into the solvent exposure of the complexes. The PPT-14–PTP1B complex displayed lower SASA values, indicating that it had a more compact structure with reduced solvent exposure. Conversely, the SASA values for the ursolic acid–PTP1B complex were comparable to those of the apo protein, suggesting similar solvent accessibility. The MD simulations of PPT-14 binding to PTP1B reveal significant intermolecular interactions that contribute to stability and potential efficacy. The formation of several hydrogen bonds with key amino acid residues, including Tyr20, Gln21, Phe29, Val48, Ser216, Arg254, Gln262, Ala264, and Lys262, suggests strong and specific interactions that likely enhance the binding affinity and specificity of PPT-14 for the PTP1B enzyme (Figure 5A). Additionally, hydrophobic interactions with residues such as Tyr46, Val49, and Ile219 further stabilize the complex, reinforcing the favorable binding environment. These interactions indicate that PPT-14 forms a well-integrated and stable complex with PTP1B, which is crucial for its potential therapeutic activity. In particular, the interactions between PPT-14 and key residues within the catalytic loop, identified by amino acids such as Asp48, Val49, Lys120, Ser216, Ala217, Gly218, Ile219, Arg220, and Gln262, are likely central to its binding and inhibition mechanisms. These residues play a crucial role in the enzymatic function of PTP1B, and the engagement of PPT-14 with these regions may be a key factor in its efficacy as an inhibitor [40,41,42]. These findings collectively demonstrate the high stability and favorable dynamic properties of the PPT-14–PTP1B complex, supporting its potential as a potent inhibitor. Its stability, compactness, and specific interactions with the protein highlight its effectiveness compared to the reference compounds.

AR, a member of the aldo-keto reductase superfamily of 190 enzymes, catalyzes the reduction of carbonyl-containing substrates, including sugar aldehydes and certain other biomolecules, and plays a significant role in lipid, carbohydrate, and xenobiotic metabolism [43]. Under normal conditions, glucose is primarily metabolized via the glycolytic pathway. However, in hyperglycemic states, such as diabetes mellitus, there is an increased flux of glucose through the polyol pathway. AR facilitates the first and rate-limiting step of the polyol pathway, converting glucose into sorbitol, which is subsequently oxidized to fructose by sorbitol dehydrogenase. Osmotic stress caused by sorbitol accumulation, along with redox imbalance due to NADPH depletion, leads to organ damage and cellular injury, contributing to the development of cataracts, neuropathy, nephropathy, and other diabetic complications [44]. As a result, AR inhibitors (ARIs) have emerged as a promising therapeutic approach to prevent the onset of these metabolic complications by targeting and inhibiting the first step of the polyol pathway. Several ARIs, such as alrestatin, benurestat, and epalrestat, among others, have been developed to treat diabetic complications. However, many of these were discontinued after clinical trials due to ineffectiveness or serious side effects, including fever, nausea, liver enzyme increases, rashes, toxic epidermal necrolysis, and conditions like thrombocytopenia and adult respiratory distress syndrome [45]. Epalrestat, approved in Japan in 1992, is the only ARI currently available for the treatment of diabetic neuropathy in countries such as Japan, China, and India [46]. Therefore, discovering effective and beneficial ARIs from natural sources is crucial for developing novel anti-diabetic treatments.

In this study, PPT-14 demonstrated significant AR inhibitory activity, with an IC_50_ value of 19.50 µM, which was greater than that of the positive control quercetin, which had an IC_50_ value of 3.2 µM. Considering the promising in vitro AR inhibitory activity of PPT-14, we conducted further in silico analyses, including molecular docking and MD simulations, and compared the results with epalrestat, a clinically approved ARI. PPT-14 exhibited a strong binding affinity, achieving a docking score of −9.1 kcal/mol, which outperformed epalrestat’s docking score of −9.7 kcal/mol. These findings underlined the promising potential of PPT-14 as a potent ARI, suggesting its viability for further development as a therapeutic agent in managing AR-associated diabetic complications. The MD simulations of the AR complexes provided insights into the structural stability and dynamics of different systems, including PPT-14–AR, epalrestat–AR, and apo AR. The RMSD analysis offered valuable insights into the stability of the complexes studied. The PPT-14–AR complex exhibited good stability, with only a brief deviation around 94 ns that stabilized after 115 ns, reflecting a strong and consistent interaction between the peptide and the protein (Figure 6A). In contrast, the epalrestat–AR complex displayed the largest deviations, highlighting its relatively lower stability, which may indicate weaker binding or less favorable interactions compared to the PPT-14–AR complex. The apo protein showed progressive instability after 160 ns, with RMSD values remaining elevated, emphasizing structural flexibility and the lack of stabilization in the absence of a bound ligand. These observations collectively underscored the stabilizing effect of the PPT-14 ligand binding on the AR protein, with the PPT-14–AR complex being more stable than epalrestat–AR complex. The RMSF analysis revealed key regions of flexibility and fluctuation within the protein structure upon binding with small molecules. The most significant fluctuations were observed at residues 212–227 across all complexes, suggesting that this region is particularly dynamic and may play a critical role in accommodating ligand binding or structural rearrangements (Figure 6B). Additionally, residues 285–310 exhibited the highest fluctuations in the apo protein compared to the ligand-bound complexes, indicating that ligand binding contributes to the stabilization of this region. These results highlighted the stabilizing effect of small-molecule interactions on specific protein regions and the inherent flexibility of the apo form. The R_g_ analysis presented the structural compactness and stability of protein complexes. The PPT-14–AR complexes exhibited more stable R_g_ profiles, indicating that ligand binding helps to maintain the protein’s structural integrity and compactness throughout the simulation (Figure 6C). In contrast, the epalrestat–AR complex had R_g_ values comparable to those of the apo protein up to 164 ns, suggesting similar levels of structural flexibility during this period. However, the subsequent decrease in R_g_ values for the epalrestat–AR complex indicated the gradual compaction of the protein structure, potentially due to conformational rearrangements or partial destabilization. These findings underscored the stabilizing effect of PPT-14, while the behavior of the epalrestat–AR complex suggested that there were weaker or less consistent interactions with the protein. The SASA profiles revealed important details about the exposure of the protein’s surface to the solvent and its structural dynamics. The PPT-14–AR complex exhibited SASA values similar to the apo protein until 106 ns, after which the values decreased, signifying a gradual reduction in surface exposure likely due to conformational stabilization induced by the peptide (Figure 6D). However, the epalrestat–AR complex closely mirrored the SASA profile of the apo protein up to 164 ns, reflecting comparable structural behavior during this period. The absence of significant reductions in the SASA suggested that epalrestat binding does not induce substantial conformational changes or stabilization. These observations highlighted the superior stabilizing effects of PPT-14 compared to epalrestat and the apo form. The analysis of intermolecular interactions indicated key residues involved in stabilizing the complexes with AR. Hydrogen bonds were observed at critical residues such as Thr19, Trp20, Lys77, His110, Ser159, Asn160, Ser210, Ile260, and Lys262, indicating strong polar interactions that contribute significantly to the stability of the protein–ligand complexes (Figure 7A). These hydrogen bonds suggested the existence of robust and specific binding at multiple interaction sites across the protein structure. Additionally, hydrophobic interactions were identified at residues Trp20, His110, Tyr209, Pro215, and Ile260, further reinforcing the stability of the complexes. These non-polar interactions were essential for maintaining the structural integrity of the binding pocket and enhancing ligand affinity. Overall, the presence of both hydrogen and hydrophobic bonds emphasized the diverse and complementary nature of interactions stabilizing the PPT-14–AR and epalrestat–AR complexes. In particular, the interactions between PPT-14 and highly conserved residues within the catalytic loop of AR, identified by amino acids such as Lys77, His110, Ser159, Asn160, Tyr209, Ser210, and Lys262, are likely central to its binding and inhibition mechanism [47,48]. These findings suggest that these residues play a pivotal role in ligand recognition and binding, with potential implications for designing more effective ARIs.

The pharmacokinetic and toxicity profile analyses of PPT-14 reveal several notable characteristics (Table 3 and Table 4). PPT-14 exhibits poor oral bioavailability, as evidenced by its low Caco2 permeability (log Papp = 0.064) and intestinal absorption (30.46%), while its inability to penetrate the BBB suggests that it is unlikely to exert direct effects on the central nervous system. PPT-14 is neither a substrate nor an inhibitor of CYP2D6 or CYP3A4 enzymes, indicating low potential for metabolic interactions, which is beneficial for safety and pharmacokinetic stability. PPT-14 has been found to be non-toxic in the AMES test, suggesting that it is unlikely to be mutagenic. Additionally, the compound exhibits no hepatotoxicity and is classified as non-toxic in the Minnow toxicity test (log mM = 3.745). While the oral rat acute toxicity (LD_50_ = 2.984 mol/kg) suggests moderate toxicity, it falls within acceptable ranges for therapeutic compounds. Furthermore, the maximum tolerated dose in humans (log mg/(kg⋅day) = 0.944) indicates that PPT-14 has good tolerability. Overall, PPT-14 has a favorable safety profile with low toxicity and no significant risks of drug–drug interactions. However, its low intestinal absorption and permeability may limit its oral bioavailability, which could be addressed through formulation strategies or modifications to improve its pharmacokinetic properties. PPT-14 exhibits favorable drug-like properties based on Lipinski’s Rule of Five, with no violations, indicating that it is “druggable” (Table 4). The physicochemical characteristics reveal a molecular weight of 312.27 g/mol, which is below the 500 Da threshold, and the compound has 8 hydrogen bond acceptors and 5 hydrogen bond donors. Both of these values are within acceptable ranges for drug-like compounds. Despite being poorly lipophilic (log Po/w = 1.36), PPT-14 demonstrates excellent water solubility (log S = −1.52), making it very soluble. However, it has a PAINS alert (catechol A), which may lead to false positive results, although this does not necessarily impact its overall drug-likeness. Furthermore, PPT-14 is classified as having “lead-likeness,” suggesting it is a promising candidate for further medicinal chemistry development.

## 4. Materials and Methods

### 4.1. General Experimental Procedures

^1^H- and ^13^C-NMR spectra were measured with a JEOL JNM ECP-400 spectrometer (Tokyo, Japan) at 400 MHz for ^1^H NMR and at 100 MHz for ^13^C NMR in deteriorated water. Column chromatography was performed using silica gel 60 (70~230 mesh, Merck, Darmstadt, Germany), LiChroprep^®^ RP-18 (40~63 μm, Merck, Darmstadt, Germany), sephadex LH-20 (20~100 μm, Sigma, St. Louis, MO, USA), and Diaion HP-20 (250~850 μm, Sigma). Thin-layer chromatography (TLC) was conducted on precoated Merck Kieselgel 60 F_254_ plates (20 × 20 cm^2^, 0.25 mm) and RP-18 F_254_ plates (5 × 10 cm^2^, Merck), using 50% H_2_SO_4_ as a spray reagent.

### 4.2. Chemicals and Reagents

1.1-Dipheny-2-picrylhydrazyl (DPPH), L-ascorbic acid, and dimethyl sulfoxide (DMSO) were purchased from Sigma-Aldrich (St. Louis, MO, USA). All other chemicals and solvents were purchased from Duksan Pure Chemicals Co., E. Merck, or Fluka, unless stated otherwise.

### 4.3. Isolation of 4-O-Caffeoyl-2-C-Methyl-d-Threonic Acid (PPT-14) from A. capillaris

Dried powder of the whole plant of *A. capillaris.* was extracted with hot MeOH for 3 h. Then, the MeOH extract was suspended in distilled water and successively partitioned with dichloromethane, ethyl acetate (EtOAc), and *n*-butanol (*n*-BuOH). The *n*-BuOH fraction was further chromatographed over an HP-diaion column and eluted with 100% H_2_O, 40% MeOH, 60% MeOH, and 100% MeOH to obtain 4 subfractions (Jung et al., 2012). Among them, the 100% H_2_O fraction (65 g) was chromatographed over a silica gel column using EtOAc:MeOH:H_2_O (24:2:1), with a gradual increase in MeOH, to produce 20 subfractions (F-1 to F-20). Fraction 19 was further chromatographed using 100% H_2_O, 50% MeOH, and 100% MeOH in order to yield 3 subfractions (F-19-1–F-19-3). Repeated chromatography of F-19 (3.44 g) over silica gel, reversed-phase (RP), and sephadex columns using different solvents yielded PPT-14 (31 mg). The structure of PPT-14 was confirmed with ^1^H and ^13^C NMR spectroscopy as well as via comparison with the published literature [26,27].

### 4.4. Assay for DPPH Radical Scavenging Activity

The DPPH radical scavenging activity was evaluated using methods outlined in Blois (1958), with slight modifications [49]. Different concentrations in 160 μL of MeOH (final concentration = 320 μg/mL for the extracts, fractions, and the compounds) were added to 40 μL of a DPPH:MeOH solution (1.5 × 10^−4^ M). After gently mixing the sample and letting it stand at room temperature for 30 min, the optical density was measured at 520 nm using a VERSAmax microplate spectrophotometer (Molecular Devices). The antioxidant activity of the samples was expressed in terms of IC_50_ values (μg/mL or μM required to inhibit DPPH radical formation by 50%), which were calculated from the log–dose inhibition curve. We used L-ascorbic acid as the positive control. The IC_50_ values presented here are expressed as the mean ± the standard error of the mean of triplicate experiments.

### 4.5. Assay for α-Glucosidase-Inhibitory Activity

Enzyme inhibition studies were carried out spectrophotometrically in a 96-well microplate reader using a procedure reported by Li [50]. A 60 µL reaction mixture, containing 20 µL of 100 mM phosphate buffer (pH 6.8), 20 µL of 2.5 mM p-NPG in the buffer, and 20 µL of PPT-14 dissolved in 10% DMSO, was added to each well. The enzymatic reaction was initiated by the addition of 20 µL of 10 mM phosphate buffer (pH 6.8), containing 0.2 U/mL α-glucosidase to each well. The plate was incubated at 37 °C for 15 min, and then 80 µL of 0.2 mol/L sodium carbonate solution was added to stop the reaction. Immediately after that, the absorbance was recorded at 405 nm using a VERSA max (Molecular Devices, Sunnyvale, CA, USA) microplate reader. Control samples contained the same reaction mixture, except that an equivalent volume of phosphate buffer was added instead of the sample solution. Acarbose dissolved in 10% DMSO was used as a positive control. The inhibition (%) was calculated as (Ac − As)/Ac × 100%, where Ac was the absorbance of the control, and As was the absorbance of the sample.

### 4.6. Assay for PTP1B-Inhibitory Activity

The inhibitory activity of vicenin 2 against PTP1B was evaluated using para-nitrophenylphosphate (p-NPP) [51]. To each well in a 96-well plate (final volume 110 µL), we added 2 mM of *p*-NPP and PTP1B to a buffer containing 50 mM citrate (pH 6.0), 0.1 M NaCl, 1 mM ethylenediaminetetraacetic acid, and 1 mM dithiothreitol with or without the sample. The plate was preincubated at 37 °C for 10 min, and then 50 µL of p-NPP in buffer was added. Following incubation at 37 °C for 30 min, the reaction was terminated with the addition of 10 M NaOH. The amount of *p*-nitrophenyl produced after enzymatic dephosphorylation was estimated by measuring the absorbance at 405 nm using a VERSA max microplate reader. The nonenzymatic hydrolysis of 2 mM *p*-NPP was corrected by measuring the increase in absorbance at 405 nm obtained in the absence of PTP1B enzyme. The inhibition (%) was calculated as above. Ursolic acid was used as the positive control.

### 4.7. Assay for Human Recombinant Aldose Reductase (HRAR)-Inhibitory Activity

The HRAR-inhibitory activities were examined as described in previous work [52]. The reaction mixture was prepared as follows: we combined 100 µL of 0.15 mM nicotinamide adenine dinucleotide phosphate (NADPH), 100 µL of 10mM DL-glyceraldehyde as a substrate, and 5 µL of HRAR with the sample (final concentration of 100 µM for the test compound, dissolved in 100% DMSO) in a total volume of 1.0 mL of 100 mM sodium phosphate buffer (pH 6.2). The AR activity was determined by measuring the decrease in nicotinamide adenine dinucleotide phosphate (NADPH) absorption at 340 nm over a period of 1 min using an Ultrospec-2100pro UV–visible spectrophotometer with Microsoft™ Excel, control by SWIFT II software (Amersham Biosciences, Piscataway, NJ, USA). Quercetin, a well-known AR inhibitor (ARI), was used as a positive control. The inhibition percentage (%) was calculated as described for the assay of Renilla luciferase, except that the DA sample/min represented a reduction in absorbance for 1 min with the test samples and substrate. The HRAR-inhibitory activity of each sample was expressed in terms of the IC_50_ value (µM), which was calculated from the log–dose inhibition curve.

### 4.8. Protein Preparation and Molecular Docking in the Inhibition of PTP1B and AR

For the docking studies, the crystal structures of the protein targets PTP1B and AR were retrieved from the RCSB Protein Data Bank (PDB) (https://www.rcsb.org/structure/2fom) in a PDB format. The PTP1B protein was chosen based on its PDB ID, 1NNY, which corresponds to a structure with a molecular weight of 38.09 kDa, a resolution of 2.40 Å, and an amino acid sequence length of 321 residues. This is recognized as one of the most reliable structures developed to date [53]. Similarly, AR was selected using PDB ID 1EL3, which has a molecular weight of 37.03 kDa and a resolution of 1.70 Å, making it a common choice for in silico analyses [47]. To prepare the proteins for docking, water molecules, heteroatoms, and ligands were removed from the PDB structures. Energy minimization was subsequently performed using the Swiss PDB Viewer v4.1 software [54]. This step is crucial to ensuring that ligands bind optimally to their specific target receptors, which typically occurs at the lowest energy state. Molecular docking, a dependable and widely used virtual screening method, was employed to evaluate the binding affinities and interactions between the protein–ligand complexes. PyRx software, equipped with the AutoDock Vina algorithm, facilitated this process [55]. The docking results were analyzed, and the protein–ligand complexes were ranked based on an empirical scoring function. The structures of the ligands, epalrestat, and ursolic acid were obtained in a structured data file format from the PubChem database. Finally, the interactions within the protein–ligand complexes were visualized and captured using BIOVIA Discovery Studio Visualizer.

### 4.9. Molecular Dynamics Simulation Study

A 200 ns simulation was employed to analyze the protein–ligand complexes to determine the binding stability of the PPT-14, ursolic acid, and epalrestat. Maestro 2020.4 platforms with the OPLS4 force field were used within a Linux operating system to perform molecular dynamics (MD) simulations to evaluate the different protein–ligand complex structures [56]. Additionally, the TIP3P aqueous archetype was used to establish a predetermined volume with an orthorhombic periodic boundary box. The recommended concentration, 0.15 M salt (Na^+^ and Cl^−^), was chosen to neutralize the system. Then, the protein–ligand solvated complex was subject to energy minimization for 100 ps. Using the SHAKE method, all the hydrogen atoms were removed, and the system was heated to 300 K. Finally, snapshots of the trajectory were taken at 100 ps intervals [57].

### 4.10. Pharmacokinetic Properties and Toxicity Prediction

The movement and the consequential effects of drugs in the body and organs are generally referred to in pharmacokinetics studies as “ADME”, which denotes absorption, distribution, metabolism, and excretion. Drug discovery through the in silico process of small-molecule drug candidates’ integrity, efficacy, and druggable properties are determined through the evaluation of pharmacokinetic properties [58]. Numerous in silico tools are available for the prediction of pharmacokinetic properties. Of these tools, SwissADME is a reliable web-based server that can predict physicochemical and drug-likeness properties of small-molecule drug candidates [59]. In this research, this server was used to forecast the physicochemical and drug-likeness properties of small-molecule drug candidates. Similarly, toxicity is also an important parameter that can provide information about the drug candidates’ unusual side effects, which can cause serious harmful events and more specifically organ damage in the body. For this reason, toxicity analysis is an important and mandatory step during computer-aided drug discovery events before accepting the drug candidates for in vivo testing and clinical trials. Thus, in this study, a reliable web-based server pkCSM was utilized to predict the pharmacokinetics properties and the toxicity profile to screen the top-listed drug candidates for further evaluations [60].

## 5. Conclusions

The current study highlights the promising potential of PPT-14, a rare caffeoyl ester derived from *A. capillaris*, as a multitarget therapeutic agent for diabetes mellitus and its associated complications. While PPT-14 did not exhibit inhibitory activity against α-glucosidase, it demonstrated the significant inhibition of key enzymes involved in the pathophysiology of diabetes, including PTP1B and AR. Molecular docking and dynamics simulations provided critical insights into its strong binding affinities and stable interactions with these enzymes, underscoring its potential efficacy. Additionally, PPT-14 showed potent antioxidant activity, further highlighting its role in mitigating oxidative stress, a major contributor to diabetes progression and complications. Pharmacokinetic evaluations revealed acceptable absorption properties, favorable water solubility, and strong adherence to Lipinski’s Rule of Five, suggesting drug-like potential. Although PPT-14 displayed poor lipophilicity and limited permeability across the BBB, these characteristics reduce the likelihood of neurotoxicity. Toxicity analyses confirmed its safety profile, showing no genotoxicity, hepatotoxicity, or minnow toxicity, with an acceptable LD_50_ value of 2.984 mol/kg, supporting its suitability for further drug development. The multitarget approach of PPT-14, addressing PTP1B inhibition, AR inhibition, and oxidative stress mitigation, positions it as a promising therapeutic alternative for the management of diabetes and diabetes-associated complications. However, its low intestinal absorption and permeability warrant further investigation to enhance its bioavailability through advanced formulation strategies or structural modifications. Future in vivo and clinical studies will be critical to confirming PPT-14’s efficacy and safety, paving the way for its development as a novel therapeutic agent for diabetes management.

## Figures and Tables

**Figure 1 ijms-26-01286-f001:**
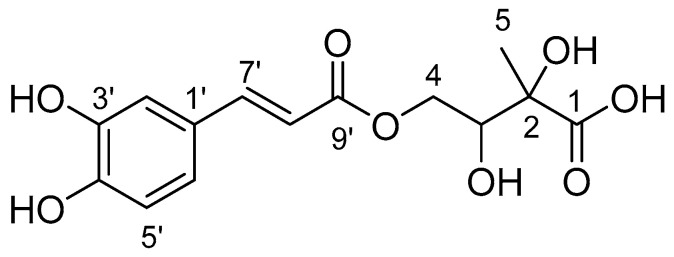
Chemical structure of 4-*O*-caffeoyl-2-*C*-methyl-d-threonic acid (PPT-14).

**Figure 2 ijms-26-01286-f002:**
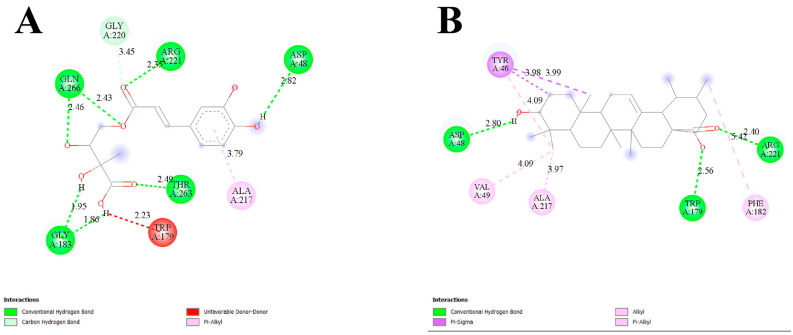
Binding interactions of PPT-14 (**A**) and ursolic acid (**B**) with the target protein PTP1B (PDB ID: 1NNY).

**Figure 3 ijms-26-01286-f003:**
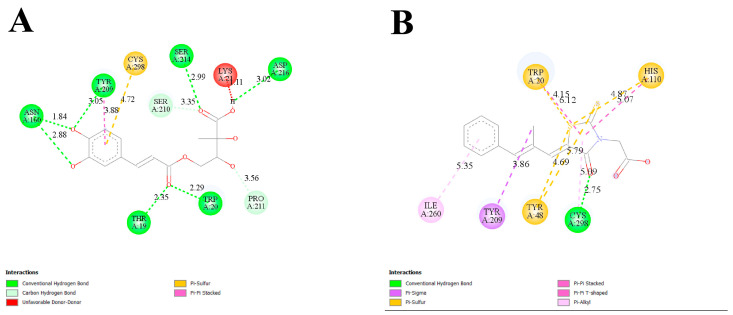
Binding interactions of PPT-14 (**A**) and epalrestat (**B**) with the target protein AR (PDB ID: 1EL3).

**Figure 4 ijms-26-01286-f004:**
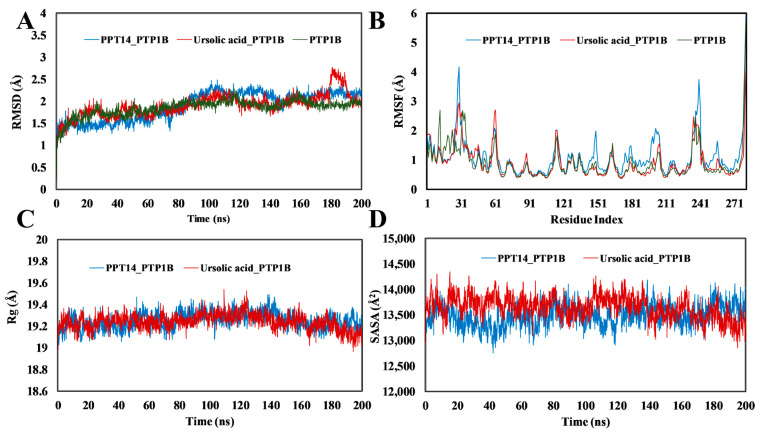
The molecular dynamics simulation of 200 ns runtime of PPT-14 protein–ligand complexes along with reference compound ursolic acid. (**A**) Root-mean-squared deviation (RMSD). (**B**) Root-mean-squared fluctuation (RMSF). (**C**) Radius of gyration (R_g_). (**D**) Solvent-accessible surface area (SASA).

**Figure 5 ijms-26-01286-f005:**
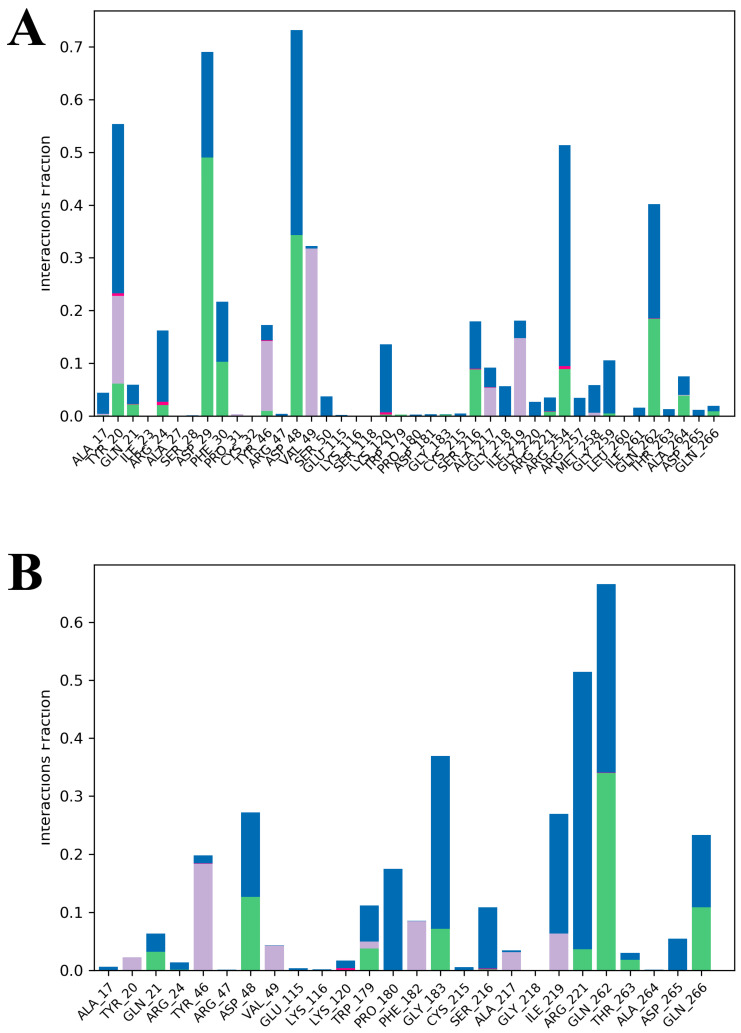
The protein–ligand interactions of PPT-14 (**A**) and ursolic acid (**B**) on the PTP1B enzyme through different types of bonds at a 200 ns simulation running time.

**Figure 6 ijms-26-01286-f006:**
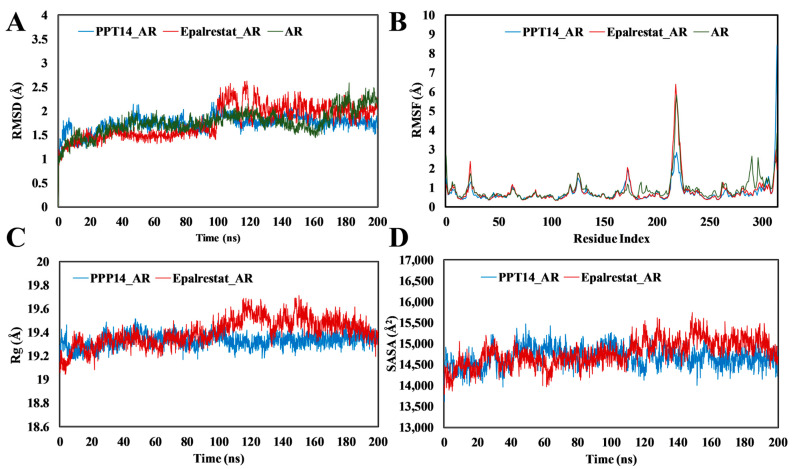
Molecular dynamics simulation of 200 ns runtime of PPT-14 protein–ligand complexes along with reference compound epalrestat. (**A**) Root-mean-squared deviation (RMSD). (**B**) Root-mean-squared fluctuation (RMSF). (**C**) Radius of gyration (Rg). (**D**) Solvent-accessible surface area (SASA).

**Figure 7 ijms-26-01286-f007:**
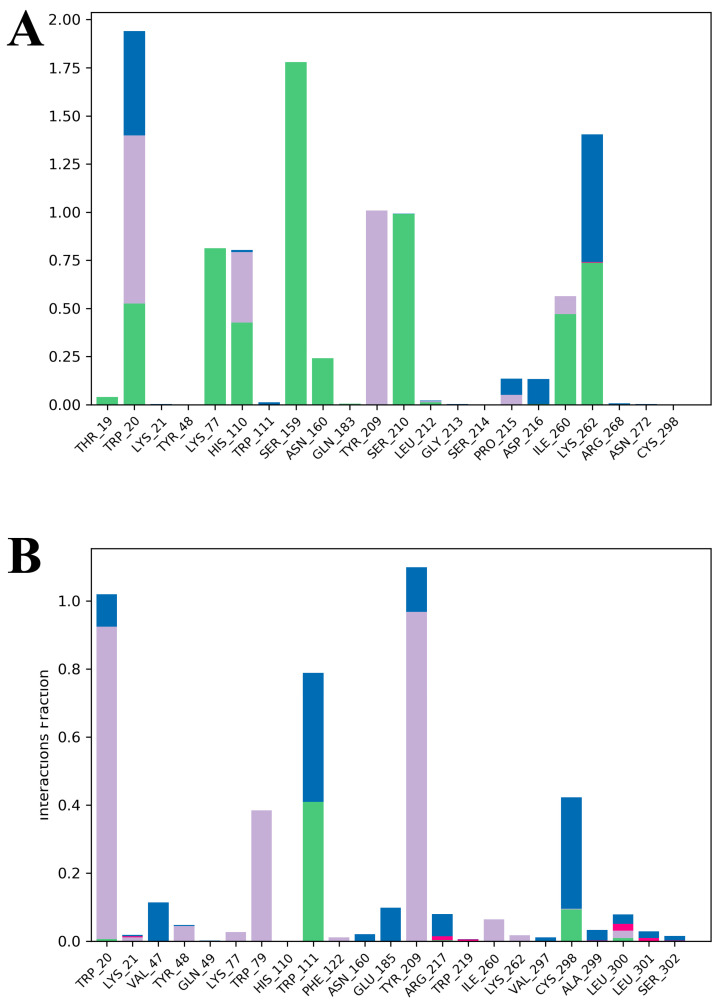
The protein–ligand interactions of PPT-14 (**A**) and epalrestat (**B**) on AR enzyme through different types of bonds at 200 ns simulation running time.

**Table 1 ijms-26-01286-t001:** PTP1B, α-glucosidase, and HRAR-inhibitory activities of PPT-14 along with DPPH free radical scavenging activity.

	IC_50_ (μM)			
Compound	PTP1B	*α*-Glucosidase	HRAR	DPPH
PPT-14	64.92 ± 0.34	>500	19.5 ± 0.51	14.46
Ursolic acid ^a^	6.11 ± 0.25	–	–	–
Acarbose ^b^	–	213.01 ± 1.22	–	–
Quercetin ^c^	–	–	3.2 ± 0.23	–
Ascorbic acid ^d^	–	–	–	31.79

^a^ positive control for PTP1B inhibition; ^b^ positive control for *α*-glucosidase inhibition; ^c^ positive control for HRAR inhibition; ^d^ positive control for DPPH scavenging assay.

**Table 2 ijms-26-01286-t002:** Docking scores and binding interactions of PPT-14 with PTP1B (PDB ID: 1NNY) and AR (1EL3). Ursolic acid and epalrestat were used as the references for PTP1B and AR, respectively.

PDB ID	Compounds	Binding Affinity (kcal/mol)	Amino Acid Interactions	
			Hydrogen Bonds	Hydrophobic Bonds
1NNY	PPT-14	7.0	Asp48 (2.82), Gly183 (1.86), Gly183 (1.95), Arg221 (2.35), Thr263 (2.49), Gln266 (2.43), Gln266 (2.46)	Trp179 (2.23) Gly220 (3.45), Ala217 (3.79)
1NNY	Ursolic acid	8.7	Asp48 (2.80), Trp179 (2.56), Arg221(2.40)	Tyr46 (3.98), Tyr46 (3.99), Tyr46 (4.09), Val49 (4.09), Phe182 (5.42), Ala217 (3.97)
1EL3	PPT-14	9.1	Thr19 (2.35), Trp20 (2.29), Asn160 (1.84), Asn160 (2.88), Tyr 209 (3.05), Ser214 (2.99), Asp216 (3.02)	Lys21 (1.11), Tyr209 (3.88), Ser210 (3.35), Pro211 (3.56), Cys298 (4.72)
1EL3	Epalrestat	9.6	Cys298 (2.75)	Trp20 (4.15), Trp20 (6.14), Tyr48 (4.69), Trp48 (5.79), His110 (4.87), His110 (5.07), Tyr209 (3.86), Ile260 (5.35), Cys298 (5.09)

**Table 3 ijms-26-01286-t003:** The analysis of the pharmacokinetics and toxicity profiles of PPT-14 isolated from *A. capillaris*.

Model Name	Unit	PPT-14
Caco2 permeability	Numeric (log Papp in 10^−6^ cm/s)	0.064
Intestinal absorption (human)	Numeric (% Absorbed)	30.462
BBB permeability	Numeric (log BB)	−1.683
CNS permeability	Numeric (log PS)	−4.675
CYP2D6 substrate	Categorical (Yes/No)	No
CYP3A4 substrate	Categorical (Yes/No)	No
CYP2D6 inhibitor	Categorical (Yes/No)	No
CYP3A4 inhibitor	Categorical (Yes/No)	No
AMES toxicity	Categorical (Yes/No)	No
Max. tolerated dose (human)	Numeric (log mg/(kg⋅day)	0.944
Oral rat acute toxicity (LD_50_)	Numeric (mol/kg)	2.984
Hepatotoxicity	Categorical (Yes/No)	No
Minnow toxicity	Numeric (log mM)	3.745

**Table 4 ijms-26-01286-t004:** The analysis of the drug-likeness profiles and physicochemical properties of PPT-14 isolated from *A. capillaris*.

Properties	Model Name	PPT-14
Physicochemical	Molecular weight	312.27 g/mol
	Num. H-bond acceptors	8
	Num. H-bond donors	5
Lipophilicity	Log Po/w (iLOGP)	1.36
Water Solubility	Log S (ESOL)	−1.52
	Class	Very soluble
Drug-likeness	Lipinski	Yes; 0 violation
Medicinal Chemistry	PAINS	1 alert: catechol_A
	Lead-likeness	Yes

## Data Availability

The datasets presented in this study can be found upon request.

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
