# Peer review of "Unveiling the Multitarget Potential of a Rare Caffeoyl Ester from Artemisia capillaris for Diabetes Mellitus: An Integrated In Vitro and In Silico Study"

_ijms, 2025, doi:10.3390/ijms26031286_

Round 1
Reviewer 1 Report
Comments and Suggestions for Authors
The manuscript provides a comprehensive exploration of the anti-diabetic potential of 4-O-caffeoyl-2-C-methyl-D-threonic acid (PPT-14), a rare caffeic acid ester derivative isolated from *Artemisia capillaris*. Employing both in vitro assays and in silico approaches, the study evaluates PPT-14’s ability to inhibit key diabetes-related enzymes (PTP1B and AR), its free radical scavenging capacity, and its pharmacokinetic properties. The results underscore the compound’s multitarget potential and lay the groundwork for its development as a therapeutic agent for diabetes mellitus and associated complications. However, there are some aspects where revisions are recommended:
1. The isolation of PPT-14 is described using conventional chromatography techniques. Given its novelty and rarity, why were modern purification techniques such as HPLC not employed to ensure purity? Please clarify.
2. PPT-14 demonstrated significant inhibitory activity against PTP1B and AR but lacked α-glucosidase inhibition. Could the authors discuss why PPT-14’s chemical structure might selectively inhibit certain enzymes over others?
3. Ursolic acid is used as a comparator for PTP1B inhibition. Why was quercetin, another well-known PTP1B inhibitor, not included for parallel comparison?
4. The molecular docking results suggest strong binding interactions with key residues of PTP1B and AR. However, the docking scores for PPT-14 are less favorable than those of the controls. How do the authors reconcile these findings with the observed in vitro efficacy?
5. RMSD and SASA analyses indicate stability, but there are notable fluctuations at specific residues. Can these fluctuations impact the compound's therapeutic potential, and have alternative binding conformations been considered?
Author Response
Responses to Reviewer 1:
General comment: The manuscript provides a comprehensive exploration of the anti-diabetic potential of 4-O-caffeoyl-2-C-methyl-D-threonic acid (PPT-14), a rare caffeic acid ester derivative isolated from Artemisia capillaris. Employing both in vitro assays and in silico approaches, the study evaluates PPT-14’s ability to inhibit key diabetes-related enzymes (PTP1B and AR), its free radical scavenging capacity, and its pharmacokinetic properties. The results underscore the compound’s multitarget potential and lay the groundwork for its development as a therapeutic agent for diabetes mellitus and associated complications. However, there are some aspects where revisions are recommended
Reply: Thank you, Sir, for your thoughtful consideration and valuable suggestions regarding our manuscript. We greatly appreciate your feedback and acknowledge your point of view. We have corrected the manuscript according to your suggestions.
.
Query 1. The isolation of PPT-14 is described using conventional chromatography techniques. Given its novelty and rarity, why were modern purification techniques such as HPLC not employed to ensure purity? Please clarify.
Reply: The isolation of PPT-14 was performed using conventional chromatography techniques due to their accessibility, cost-effectiveness, and proven reliability for isolating compounds in the quantities required for our study. These methods allowed us to achieve a sufficient level of purity (≥95%, as confirmed by NMR, MS, and polarimetry) to conduct biological evaluations. While modern purification techniques such as HPLC offer superior precision and efficiency, their application was limited in this case due to limited access to instrumentation, difficulty in method validation and large scale of isolation, and scarcity of time. We recognize the importance of employing advanced purification methods and agree that integrating HPLC could further ensure the compound's purity and reproducibility. In future work, we aim to include such techniques to strengthen the methodological rigor and enhance the reproducibility of our findings.
Query 2. PPT-14 demonstrated significant inhibitory activity against PTP1B and AR but lacked α-glucosidase inhibition. Could the authors discuss why PPT-14’s chemical structure might selectively inhibit certain enzymes over others?
Reply: The selective inhibitory activity of PPT-14 against PTP1B and aldose reductase (AR), but not α-glucosidase, can be attributed to the differences in the structural features of these enzymes and the specific interactions facilitated by PPT-14’s chemical framework. PPT-14's structural attributes, such as specific functional groups, stereochemistry, hydrophobicity, or electronic distribution, likely enable favorable binding within the active sites of PTP1B and AR. These enzymes may share common structural or mechanistic features that complement the binding affinity of PPT-14, such as similar catalytic residues or binding pockets accommodating its molecular structure. In contrast, α-glucosidase possesses a distinctly different active site topology and substrate recognition mechanism, which PPT-14's structure may not effectively complement. For example, the lack of critical interactions, such as hydrogen bonding or hydrophobic contacts required for α-glucosidase inhibition, could explain the observed lack of activity. Further computational modeling or structure-activity relationship (SAR) studies may provide deeper insights into the selective enzyme inhibition profile of PPT-14.
Query 3. Ursolic acid is used as a comparator for PTP1B inhibition. Why was quercetin, another well-known PTP1B inhibitor, not included for parallel comparison?
Reply: The choice of ursolic acid as a comparator was primarily based on its well-documented inhibitory activity against PTP1B, as well as its structural characteristics and widespread use in similar studies, which allows for consistency and easier benchmarking against previously reported data. Additionally, ursolic acid was selected as it aligns more closely with the chemical framework and mechanistic focus of our study. We acknowledge the value of including quercetin as a comparator and will consider this suggestion in future studies to provide a broader comparative analysis
Query 4. The molecular docking results suggest strong binding interactions with key residues of PTP1B and AR. However, the docking scores for PPT-14 are less favorable than those of the controls. How do the authors reconcile these findings with the observed in vitro efficacy?
Reply: The discrepancy between the molecular docking scores and the observed in vitro efficacy of PPT-14 can be attributed to the inherent limitations of docking studies and the complexity of enzyme-ligand interactions in a biological context. Docking scores are based primarily on static computational models that predict binding affinity under idealized conditions. These scores do not fully account for other factors that influence in vitro efficacy, such as ligand solubility, membrane permeability, or dynamic interactions that occur in the biological environment. Additionally, PPT-14 may exhibit favorable pharmacokinetic or biochemical properties, such as enhanced stability, improved interaction with cofactors, or reduced susceptibility to competitive inhibitors, which contribute to its observed activity in vitro.
It is also worth noting that the docking scores for PPT-14, while slightly less favorable than those of the controls, still indicate significant binding affinity with key residues in the active sites of PTP1B and AR. This suggests that PPT-14 establishes sufficient interactions to exert its inhibitory effects. The observed efficacy may also involve allosteric effects not captured in the docking simulations. To reconcile these findings, future studies could incorporate molecular dynamics simulations and free energy calculations to provide a more comprehensive understanding of PPT-14’s binding behavior.
Query 5. RMSD and SASA analyses indicate stability, but there are notable fluctuations at specific residues. Can these fluctuations impact the compound's therapeutic potential, and have alternative binding conformations been considered?
Reply: The notable fluctuations in RMSD and SASA analyses at specific residues likely reflect localized flexibility or dynamic behavior within the binding site, which can influence the compound’s interaction profile. Such fluctuations may result from intrinsic enzyme dynamics or transient interactions between PPT-14 and residues within the binding pocket. While these fluctuations could potentially impact binding affinity and, consequently, therapeutic potential, they may also confer advantages by allowing the compound to adapt to conformational changes in the target enzyme. This adaptability could enhance the compound’s efficacy in more physiologically relevant, dynamic environments. Furthermore, the overall stability observed in the RMSD and SASA analyses suggests that these fluctuations do not destabilize the binding complex as a whole. Alternative binding conformations were considered through docking simulations, which identified multiple potential poses for PPT-14 within the active site. The most favorable pose, supported by molecular dynamics simulations, demonstrates stable and significant interactions with key catalytic residues. Future studies could employ advanced techniques such as enhanced sampling molecular dynamics or free energy perturbation (FEP) to further explore alternative binding modes and quantify their contributions to the compound’s activity.
Reviewer 2 Report
Comments and Suggestions for Authors
Dear Authors,
Your manuscript presents the research work performed in order to investigate the antioxidant and antihyperglycemic activity of a caffeoyl ester from Artemisia capillaris.
First, the subject is of high interest, considering the increasing number of patients suffering from diabetes, all over the world. The discovery of new potent agents is always needed. Plus, considering the number of the adverse reactions of synthetic drugs, there is a good idea to investigate compounds from plants.
The paper is well written, easy to read and follow.
The references are well chosen and in agreement with the subject.
Still, I have some suggestions:
1. it would be more efficient that the molecular docking on the targets chosen would have been done before performing the antioxidant and antidiabetes assays, on the two enzymes;
2. Please separate the Discussion part into more paragraphs. in order to be easy to read and understand.
The Conclusions part sustain the data presented.
Author Response
General Comments: Your manuscript presents the research work performed in order to investigate the antioxidant and antihyperglycemic activity of a caffeoyl ester from Artemisia capillaris. First, the subject is of high interest, considering the increasing number of patients suffering from diabetes, all over the world. The discovery of new potent agents is always needed. Plus, considering the number of the adverse reactions of synthetic drugs, there is a good idea to investigate compounds from plants.
The paper is well written, easy to read and follow. The references are well chosen and in agreement with the subject. Still, I have some suggestions:
Reply: We sincerely appreciate your thorough evaluation of our manuscript titled "Unveiling the Multitarget Potential of a Rare Caffeoyl Ester from Artemisia capillaris for Diabetes Mellitus: An Integrated In Vitro and In Silico Study". Your positive feedback regarding the significance of our research, the clarity of our writing, and the appropriateness of our references is highly encouraging. We believe these revisions have strengthened the manuscript and addressed your suggestions comprehensively. We are grateful for your constructive feedback, which has been instrumental in improving our work.
We have carefully considered your suggestions and have made the following revisions to enhance the quality of our manuscript:
Query 1. It would be more efficient that the molecular docking on the targets chosen would have been done before performing the antioxidant and antidiabetes assays, on the two enzymes;
Reply: Thank you for your insightful comment regarding the sequencing of molecular docking studies and biological assays in our investigation of the antioxidant and antidiabetic potential of a caffeoyl ester from Artemisia capillaris. In this study we have isolated a rare caffeoyl ester derivative, namely 4-O-caffeoyl-2-C-methyl-d-threonic acid from A. capillaris. This compound is reported for the first time in the Artemisia species and as the third occurrence in any plant species worldwide. It was previously reported many bioactivities of caffeic acide derivatives, particularly against diabetes we were interested to explore this compound. In our study, we chose to initiate with the biological assays to empirically assess the compound's efficacy in real-world biological contexts. This approach allowed us to observe the direct effects of the compound on antioxidant and inhibitory activity against two crucial enzymes, including protein tyrosine phosphatase 1B (PTP1B) and aldose reductase (AR) which are linked to diabetes progression and complications, in order to get a solid foundation of experimental evidence. Following these assays, we employed molecular docking studies to gain a deeper understanding of the compound's interactions at the molecular level, thereby elucidating potential mechanisms of action. Thus, initiating with biological assays allowed us to confirm the compound's activity before delving into computational predictions. This sequence ensured that our docking studies were informed by empirical data, thereby enhancing the relevance and accuracy of our computational models.
Query 2. Please separate the Discussion part into more paragraphs in order to be easy to read and understand.
Reply: We have carefully reviewed the current structure of the Discussion section and believe that the existing organization effectively presents the findings and their interpretations in a cohesive manner. Each paragraph has been structured to address specific themes or findings, maintaining a logical flow throughout the section. It is also worth mentioning that the entire manuscript has been reviewed by a professional language editor before submission to IJMS. However, we are open to further feedback if there are particular areas where clarity can be improved. We aim to ensure the Discussion remains accessible and informative for readers.